# Hyperactive immature state and differential CXCR2 expression of neutrophils in severe COVID-19

Christopher M Rice[1], Philip Lewis[2], Fernando M Ponce-Garcia[1], Willem Gibbs[1], Sarah Groves[1], Drinalda Cela[1],
Fergus Hamilton[3,4], David Arnold[3], Catherine Hyams[1,3], Elizabeth Oliver[1], Rachael Barr[1], Anu Goenka[1],
Andrew Davidson[1], Linda Wooldridge[5], Adam Finn[1], Laura Rivino[1], Borko Amulic[1]

Neutrophils are vital in defence against pathogens, but excessive neutrophil activity can lead to tissue damage and promote acute respiratory distress syndrome. COVID-19 is associated with systemic expansion of immature neutrophils, but the functional consequences of this shift to immaturity are not understood. We used flow cytometry to investigate activity and phenotypic diversity of circulating neutrophils in acute and convalescent COVID-19 patients. First, we demonstrate hyperactivation of immature CD10⁻ subpopulations in severe disease, with elevated markers of secondary granule release. Partially activated immature neutrophils were detectable 12 wk post-hospitalisation, indicating long term myeloid dysregulation in convalescent COVID-19 patients. Second, we demonstrate that neutrophils from moderately ill patients down-regulate the chemokine receptor CXCR2, whereas neutrophils from severely ill individuals fail to do so, suggesting an altered ability for organ trafficking and a potential mechanism for induction of disease tolerance. CD10⁻ and CXCR2^hi neutrophil subpopulations were enriched in severe disease and may represent prognostic biomarkers for the identification of individuals at high risk of progressing to severe COVID-19.

## Introduction

Severe acute respiratory syndrome coronavirus 2 (SARS-Cov-2), the causative agent of COVID-19, is a single-stranded RNA virus that is transmitted by respiratory droplets. The clinical spectrum of COVID-19 is wide, ranging from a paucisymptomatic self-limiting upper airway infection to a lower respiratory tract infection (1), associated with acute respiratory distress syndrome (ARDS). The hyper-inflammatory state in ARDS impairs lung function and is the cause of death in 70% of such patients (1, 2). Other systemic manifestations of COVID-19 include thromboembolism (3), acute kidney damage (4), and cardiac injury (5).

Excessive host inflammatory responses are recognized as central to the pathogenesis of severe COVID-19. Viral proliferation induces type I interferons and proinflammatory cytokines (6), which may be protective early in infection but lead to cytokine storm and tissue damage when produced excessively (7). Immunosuppressive therapies, such as the glucocorticoid dexamethasone, significantly reduce mortality in hospitalized patients receiving respiratory support (8). Dexamethasone therapy, however, is not successful in all patients, and the mechanistic basis for this failure is not understood.

Neutrophils are abundant myeloid cells that are vital for defence against pathogens, acting to suppress bacterial and fungal dissemination. Their antimicrobial response includes the production of reactive oxygen species (ROS), degranulation of proteases, and release of extracellular chromatin in the form of neutrophil extracellular traps (NETs) (9). The role of neutrophils in the anti-viral response is less clear (10). Importantly, excessive or dysregulated neutrophil activity can promote immunopathology and is thought to be a key feature of ARDS (11).

Despite improved insight into the inflammatory injury in SARS-CoV-2 infection, our understanding of the neutrophil contribution remains incomplete. Transcriptomic and proteomic analyses of COVID-19 patient peripheral blood (12, 13, 14, 15, 16) and bronchoalveolar lavage fluid (17, 18) reveal a disordered myeloid response. Peripheral neutrophil activation predicts clinical outcome and is strongly associated with mortality (12, 13, 15). Similarly, circulating neutrophil counts are elevated in severely ill patients, and higher neutrophil-to-lymphocyte ratios are associated with poor prognosis (19, 20, 21). A GWAS identified elevated neutrophil count as a causal trait for hyperinflammation in critical COVID-19 (22 Preprint). Neutrophil chemoattractants, such as CXCL1 and CXCL8 (IL-8) (23, 24), as well as transmigrated, infiltrating neutrophils (23, 25, 26, 27, 28) are detected in the alveolar space of infected lungs. Neutrophil-soluble mediators, such as elastase and calprotectin, accumulate in plasma of patients with severe disease (29, 30). Furthermore, NETs are elevated in the plasma of severe patients and correlate with lung failure and immunothrombosis (25, 31, 32, 33 Preprint), whereas isolated neutrophils from COVID-19 patients

[1]School of Cellular and Molecular Medicine, Faculty of Life Sciences, University of Bristol, Bristol, UK  [2]University of Bristol Proteomics Facility, Faculty of Life Sciences, University of Bristol, Bristol, UK  [3]Academic Respiratory Unit, Bristol Medical School, University of Bristol, Southmead Hospital, Bristol, UK  [4]MRC Integrative Epidemiology Unit, University of Bristol, Bristol, UK  [5]Bristol Veterinary School, Faculty of Health Sciences, University of Bristol, Bristol, UK

Correspondence: borko.amulic@bristol.ac.uk; christopher.rice@bristol.ac.uk

display elevated rates of both ROS production (34) and NETosis (35, 36). Several studies have also reported expansion of low-density (37) and immature (CD10$^{low}$) neutrophils (30, 38, 39 *Preprint*, 40), whose role in disease remains unclear. Collectively, these observations suggest that neutrophil activation plays a significant role in disease progression and eventual fatality of COVID-19.

We performed a detailed multidimensional flow cytometry analysis and proteomic characterization of circulating neutrophils in patients with COVID-19. We discerned neutrophils with high discrimination from closely related leukocyte subsets and examined receptors associated with activation, maturation, and trafficking. COVID-19 patients exhibit a pattern of systemic neutrophil activation and increased neutrophil heterogeneity. Hyperactive immature states and maintenance of CXCR2 are prognostic of poor patient outcome. Our study highlights the importance of neutrophil developmental state when considering pathogenic function in COVID-19 and identifies potential biomarkers of severe disease.

# Results

### Immature circulating neutrophils are enriched in moderate and severe COVID-19

To assess phenotypic changes in neutrophil populations, we analysed fresh peripheral blood from hospitalised acute COVID-19 patients (n = 34; median days since symptom onset = 13 ± 5.29; median days since hospital admission = 4 ± 3.99; Table 1), convalescent individuals (n = 38; days since symptom onset median = 76.5 ± 30.7), and healthy controls (n = 20) (Table 2). Patients were stratified by disease severity (41), with severe disease defined by requirement for ventilation and/or intensive care admission, moderate disease defined as requirement for supplementary oxygen without intensive care, and mild disease requiring no oxygen supplementation and no intensive care. Stratification of patients was representative of status at time of sample acquisition, except when analysing survival. Patients with severe disease were further stratified into surviving and deceased groups.

Whole blood was labelled using a panel of antibodies targeting surface markers designed to identify neutrophils with a high degree of discrimination and assess phenotypic changes. We identified neutrophils as CD14⁻, CD15⁺, and CD125⁻ (Fig S1A), thereby avoiding the commonly used marker CD16 (FcYRIII), which varies in expression with neutrophil maturity (42) and is the target of surface proteases (30, 43). To ensure CD15⁺ CD16⁻ eosinophils were not incorrectly identified as neutrophils, we excluded these based on of IL-5R (CD125) expression (Fig S1A). Our gating strategy was further validated by comparing staining of purified neutrophils with both our flow cytometry antibody panel and Giemsa nuclear dye to confirm typical neutrophil nuclear morphology (Fig S1B and C). Despite a trend towards an increased neutrophil count (Fig S1D) and elevated neutrophil-to-lymphocyte ratio (Fig S1E), in severely

**Table 1. Cumulative clinical data of COVID-19 patients.**

| COVID-19 patients | Mild (n = 3) | Moderate (n = 14) | Severe (n = 18) |
|---|---|---|---|
| Demographic and clinical data | | | |
| Age, years median ± SD | 78 ± 6.48 | 52.5 ± 10.59 | 65 ± 13.31 |
| Sex, M/F | 0/3 | 9/5 | 14/4 |
| Intensive/high care admission (%) | 0% | 7% | 83% |
| Deceased within 33 d of admission | 0/3 | 0/14 | 6/18 |
| Peripheral blood laboratory parameters median ± SD | | | |
| Leukocyte count × 10⁹/Liters | 6.36 ± 1.4 | 6.74 ± 3.7 | 8.69 ± 5 |
| Neutrophil count × 10⁹/Liters | 4.59 ± 0.89 | 4.79 ± 3.1 | 7.35 ± 4.79 |
| Lymphocyte count × 10⁹/Liters | 1.22 ± 0.42 | 1.09 ± 0.47 | 0.95 ± 0.49 |
| Neutrophil-to-lymphocyte ratio | 3.76 ± 1.22 | 5.02 ± 4.27 | 5.91 ± 8.03 |
| Serum D-dimer | a | 0.59 ± 6.03 | 0.56 ± 0.12 |
| Serum CRP (mg/dl) | 42 ± 91.68 | 62.5 ± 45.3 | 131 ± 109.4 |
| Comorbidities | | | |
| Diabetes | 1/3 | 3/14 | 3/18 |
| Heart disease | 2/3 | 1/14 | 6/18 |
| Chronic kidney disease | 1/3 | 0/14 | 0/18 |
| Sample timeline median ± SD | | | |
| Days since symptom onset | b | 9.5 ± 4.45 | 14 ± 5.07 |
| Days since hospitalisation | 5 ± 5.1 | 2 ± 1.87 | 5.5 ± 4.07 |

[a]Data not available.
[b]Patients did not report a symptom onset date.
Statistical analysis was performed as follows: leukocyte count, neutrophil count, lymphocyte count, neutrophil-to-lymphocyte count, D-dimer, and CRP were analysed by one-way ANOVA with Tukey's multiple comparisons. No parameter tested was significant.

**Table 2. Convalescent and healthy control clinical information.**

| Healthy controls and convalescent donors | Healthy controls (n = 20) | Convalescent (n = 38) | |
|---|---|---|---|
| | | Outpatients (n = 22) | Hospitalised (n = 16) |
| Demographic and clinical data | | | |
| Age, years median ± SD | 45.7 ± 11.63 | 31 ± 13.49 | 60.5 ± 11.97 |
| Sex, M/F | 10/10 | 7/15 | 13/3 |
| Intensive/high care admission (%) | n/a | n/a | 18.7% |
| Comorbidities | | | |
| Diabetes | 0/22 | 1/22 | 2/16 |
| Heart disease | 0/22 | 0/22 | 4/16 |
| Chronic kidney disease | 0/22 | 0/22 | 0/16 |
| Sample timeline median ± SD | | | |
| Days since symptom onset | n/a | 40 ± 22.29 | 91 ± 4.96 |
| Days since hospitalisation | n/a | n/a | 85 ± 4.45 |

affected patients, no significant differences in these parameters were observed in our cohort.

CD10, also known as membrane metalloendopeptidase, is a neutrophil maturity marker whose expression is absent or less abundant on immature neutrophils (44, 45). We detected reduced CD10 expression on neutrophils from moderately and severely ill COVID-19 patients compared with both healthy controls and convalescent individuals (Fig S2A). Moderate and severe groups were also enriched in CD10-negative (CD10⁻) neutrophils (Fig 1A), determined by comparison to CD10⁻ monocytes (Fig S2B). CD10 surface expression was lowest amongst patients that subsequently succumbed to the disease (non-survivors) (Fig S2C), and CD10⁻ neutrophils were significantly enriched in patients admitted to high/intensive care settings (Fig S2D), suggesting prognostic utility. We confirmed neutrophil immaturity by quantifying expression of an additional marker, CD101 (Fig S2E), which is similarly reduced or absent on immature cells (42, 46). Taken together, our results conclusively demonstrate accumulation of immature neutrophils in circulation of COVID-19 patients, consistent with other reports (14, 38, 47).

### Peripheral neutrophil activation in severe COVID-19

We assessed neutrophil activity by quantifying reduction of CD62L (L-selectin) expression, a surface receptor that is proteolytically shed upon stimulation (48). CD62L$^{low}$ neutrophils were elevated in severe COVID-19 patients, demonstrating acute peripheral activation (Fig 1B). We next analysed neutrophil degranulation and detected increased surface abundance of CD66b, a secondary granule marker (49), across all patient groups (Fig 1C). We also detected a trend towards increased abundance of CD63 (primary granule marker)-positive neutrophils in severe patients (Fig S2F), but found no difference in CD11b expression (tertiary and secretory vesicle marker) (Fig S2G). CD66b or CD63 exocytosis and loss of CD62L expression did not differ with survival status of severe patients (Fig S2H).

To further explore secondary granule release, we analysed expression of CD177, a protein that localises to both the plasma membrane and secondary granules and is up-regulated upon degranulation (50). In most of the healthy individuals, CD177 demonstrates bimodal expression, with ~50% of circulating neutrophils expressing this receptor (51) (Fig S2I). We detected significant up-regulation of CD177 expression in neutrophils from moderate and severe COVID-19 groups (Fig 1D) and a significant increase in overall percentages of CD177⁺ neutrophils (Fig S2I). Furthermore, patients who later succumbed to disease displayed a trend towards greatest expression of CD177 (Fig S2J). In summary, moderate and severe COVID-19 are associated with acute neutrophil activation and release of secondary granules.

### Severe COVID-19 patients maintain naive CXCR2 expression levels

Trafficking of neutrophils to organs, including the lung, is orchestrated by chemokines such as CXCL8 (IL-8) and CXCL1 (52), which primarily signal via the CXCR2 chemokine receptor. This G protein-coupled receptor shapes the neutrophil chemotactic response by regulating the actin cytoskeleton (53). At high concentrations of chemokines, CXCR2 is internalised by endocytosis, which has been suggested as a form of terminal activation, after which cells are no longer responsive to cognate chemokines (54). Interestingly, CXCR2 expression was significantly reduced in moderately ill patients (Fig 1E). Comparison of CXCR2 expression in these patients with particularly low CXCR2 to a fluorescence minus one control suggested moderately ill patients have minimal surface CXCR2 expression (Fig S3A). Severely ill patients, in contrast, maintained CXCR2 expression at levels similar to those found in healthy or convalescent individuals (Fig 1E), although there was no significant difference between survivors and non-survivors (Fig S3B). CXCR2 expression was also examined in patients stratified by care status, with a non-significant trend for increased CXCR2 in patients in intensive/high care settings (Fig S3C). Measurement of plasma CXCL8 replicated previously reported data (55), showing elevated CXCL8 in COVID-19 patients compared with healthy controls, with a trend for increased levels in severe patients (Fig S3D). Comparison of serum CXCL8 to surface CXCR2 expression failed to

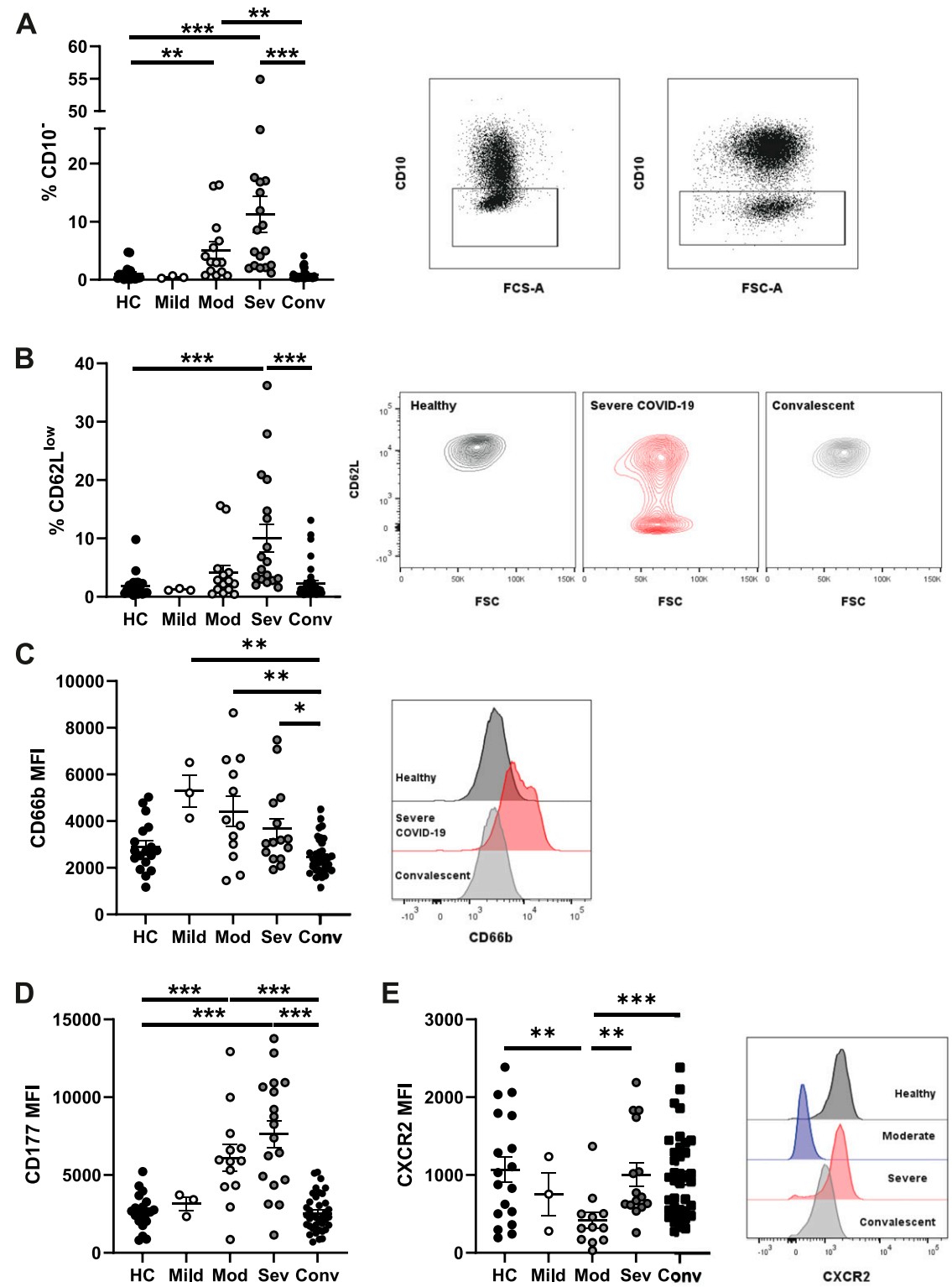

**Figure 1. Alterations to circulating neutrophil phenotype during acute SARS-CoV-2 infection.**
Peripheral blood from healthy controls, convalescent, and acute COVID-19 patients was analysed by flow cytometry. **(A)** Percentage of CD10⁻ neutrophils from stratified groups (n = HC-20, mild 3, mod 14, sev 18, and conv 38) and representative gating for CD10⁻ neutrophils from two severe patients. **(B)** Percentage of CD62L$^{low}$ neutrophils and representative contour plots of CD62L expression (n = HC-20, mild 3, mod 14, sev 18, and conv 38). **(C)** Neutrophil CD66b median fluorescent intensity and representative histogram of CD66b fluorescent intensity (n = HC-20, mild 3, mod 12, sev 15, and conv 38). **(D)** CD177 median fluorescent intensity on CD177⁺ neutrophils (n = HC-20, mild 3, mod 13, sev 18, and conv 38). **(E)** Neutrophil CXCR2 median fluorescent intensity and representative histogram of CXCR2 fluorescent intensity (n = HC-18, mild 3, mod 12,

explain the mechanism of down-regulated CXCR2 in moderately ill patients (Fig S3E). Assessment of CXCR2 transcription in a small number of moderately ill patients found reduced transcription when compared with healthy controls (Fig S3F). In addition to CXCR2, we also found a similar, but non-significant, trend towards maintenance of CXCR4 expression in severe patients (Fig S3G). In summary, neutrophils in patients with moderate disease down-regulate surface CXCR2 expression, potentially limiting their capacity to migrate to inflamed tissues. In contrast, neutrophils in patients with severe disease are characterised by a failure to down-regulate CXCR2 expression, indicating a maintained capacity for trafficking to organs such as the lungs.

### Neutrophils from patients with severe COVID-19 up-regulate pathways for chemotaxis and protein synthesis

To further explore neutrophil activation and chemotactic capacity, we compared the proteomes of circulating neutrophils from severely ill COVID-19 patients (n = 3; % CD10⁻ = 9.4–17.6%) and healthy controls (n = 4) (Table S1). We detected a total of 3,316 proteins (Fig S4A), including 466 proteins with significantly altered expression ($P < 0.05$) (Fig S4B), of which 75 also passed a threshold of twofold difference. Principle component analysis (PCA) demonstrated heterogeneity amongst the severe patients, which was explained by patient outcome (Fig S4C); however, because of the small sample size, subsequent analysis was performed on combined patient samples.

The most strongly induced significant protein was interferon-induced protein with tetratricopeptide repeats (IFIT) 5 (fold change = 7.65, $P = 0.021$) (Fig S4A and D), consistent with previous reports of a type I interferon signature in neutrophils from COVID-19 patients (47, 56, 57, 58). Indeed, interrogation of significantly altered proteins using the Interferome database (59) confirmed active type I and II IFN signalling in severe COVID-19 patients (28), despite dexamethasone administration (Fig S4E).

We identified changes in numerous proteins related to neutrophil function. These included reductions in components of the NADPH oxidase complex (neutrophil cytosol factor [NCF 1 and 2] and cytochrome b 558 subunit $\beta$ [CYBB]) (Fig S4F), suggesting reduced ability to produce ROS and potentially contributing to the increased incidence of fungal and bacterial co-infections in steroid-treated COVID-19 patients (60). Conversely, proteins associated with neutrophil granules such as azurocidin (AZU1), cathepsin (CTS) D and G, and proteinase (PRTN) 3 were elevated in patients, suggesting neutrophils may instead be attuned to degranulation (Fig S4F).

In addition to these neutrophil-specific proteins, ingenuity pathway analysis (IPA) identified general pathways related to chemotaxis, such as "signalling by Rho family GTPases," "RhoA signalling," and "regulation of actin-based motility by Rho," as significantly enriched in patient neutrophils (Fig S4G), which aligns with maintained expression of CXCR2 (Fig 1E). IPA also detected enrichment of RNA processing and translation pathways (cleavage and polyadenylation of pre-mRNA and eIF2 signalling) (Fig S4G

and H), suggesting elevated de novo protein synthesis. In summary, proteomic changes in patient neutrophils highlight increased chemotaxis and protein translation, which is consistent with detection of activation by flow cytometry.

### CD10⁻ immature neutrophils are hyperactivated

We next used flow cytometry data to (i) ask whether the highly enriched immature neutrophils differ phenotypically from their mature counterparts and (ii) identify subpopulations that are biomarkers of severe disease. We combined data from healthy controls, convalescent, and COVID-19 patients and performed uniform manifold approximation and projection (UMAP) (Fig 2A), a form of dimensionality reduction that generates two-dimensional representations of multiple surface markers and allows for interrogation of cellular heterogeneity (61 Preprint). Distribution of healthy controls, convalescent, and COVID-19 patients within the UMAP plot demonstrated that these three groups occupy relatively distinct spaces (Fig 2B). Interestingly, neutrophils from convalescent patients (median days since symptom onset = 76.5 ± 30.7) were characterised by an intermediate distribution between healthy controls and acute patients, suggesting long term perturbations to the myeloid compartment.

To examine the functional state of immature neutrophils, we overlayed surface marker expression on UMAP plots, which demonstrated distinct, alternatively activated clusters (Fig 2C). CD10⁻ cells were identified as a discrete population (marked by asterisk), that was simultaneously CD66b^high and CD62L^low (Fig 2C). This cluster was also identified as CXCR2^low, suggesting it does not account for maintenance of CXCR2 in severe patients (Fig S5A). We conclude that peripheral CD10⁻ cells are hyperactive, engaging in increased secondary granule exocytosis and surface protease activation.

To validate our UMAP clustering and to unbiasedly identify clusters associated with severe disease, we combined UMAP clustering with X-shift analysis (62), an algorithm which automatically identifies cell populations and selects the optimal number of clusters. Using this approach, 25 clusters were identified, which were then overlayed onto UMAP plots (Fig 2D and E). Distribution of X-Shift clusters between stratified groups supports our observation of distinct neutrophil phenotypes in different disease states (Fig 2F). This approach also confirmed our manual gating: ranking clusters by CD10 expression identified the CD10⁻ immature clusters (Fig 2C) as two clusters (clusters 23 and 24), separated by the bimodal expression of CD177 (Figs 2G and S5B). Examination of surface expression in these clusters identifies these as possessing the greatest CD66b expression and the lowest CD62L expression among all detected clusters (Fig 2E and G). These clusters were significantly enriched in severe and moderate COVID-19, demonstrating they are a feature of worsening disease (Fig 2H). Taken together, our analysis identifies hyperactive CD10⁻ neutrophil subpopulations as potential biomarkers of severe COVID-19.

---

sev 15, and conv 38). Statistical tests were performed as outlined below. **(A, B)** Data were analysed by the Kruskal–Wallis test, with Dunn's multiple comparisons displayed on the graph. **(C, D, E)** Data were log transformed and analysed by one-way ANOVAs with Tukey's multiple comparisons displayed on the graph. HC, healthy controls; Mod, moderate; Sev, severe; Conv, convalescent. Values represent mean ± SD.

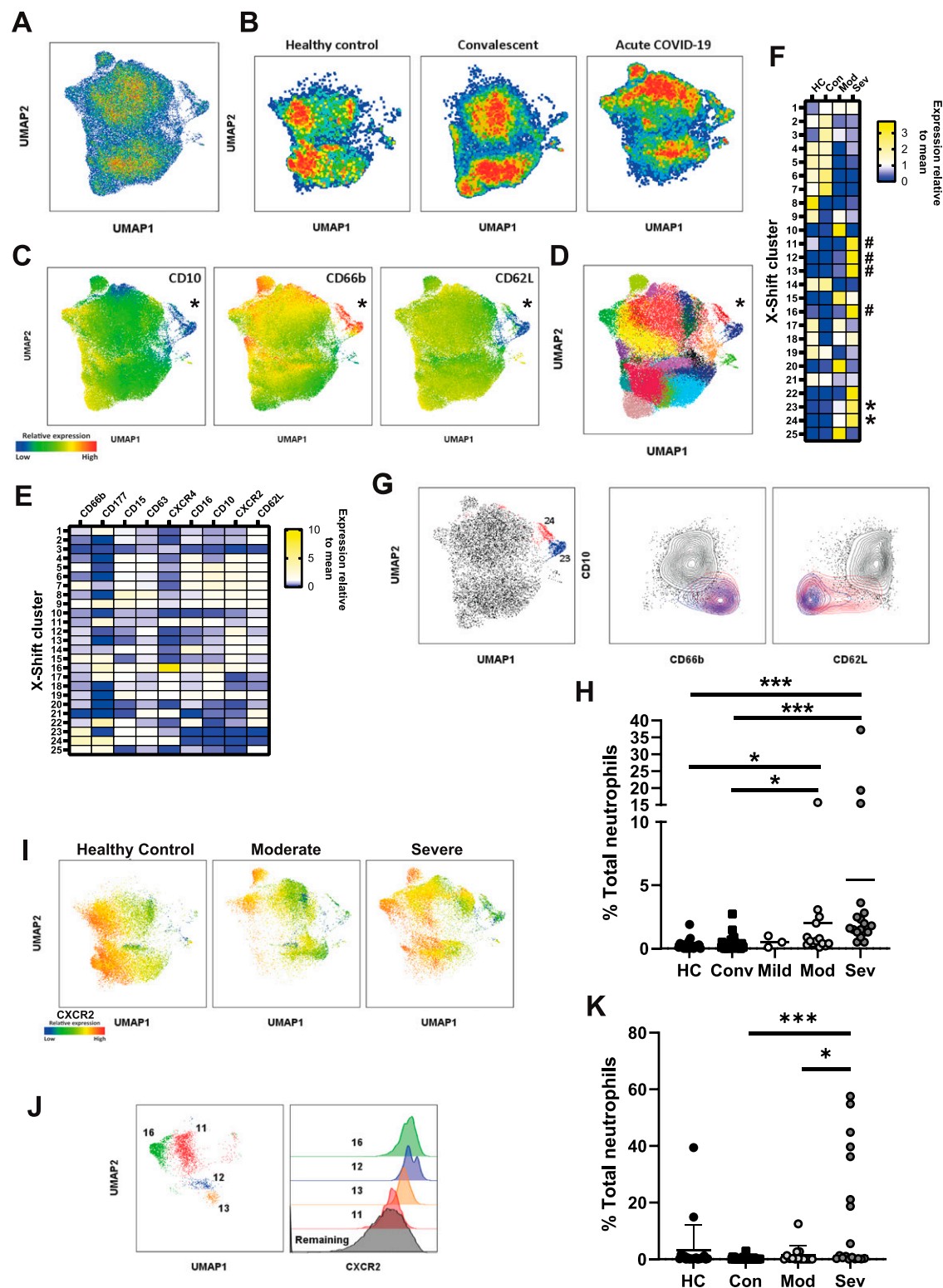

Figure 2. Immature neutrophils in severe COVID-19 are hyperactive.
(A) UMAP clustering of circulating neutrophils combined from 90 individual donors. (B) Pseudo-colour plots demonstrate distribution of healthy controls, convalescent donors, and acute COVID-19 patients within UMAP plot. (C) Expression of CD10, CD66b, and CD62L overlayed on UMAP plot with population of interest marked with an asterisk. (D) X-shift clusters overlayed onto UMAP plots. (E) Median fluorescent intensity of surface markers from 25 automatically generated X-shift clusters. (F) Relative distribution of X-Shift clusters between stratified groups. * = hyperactive immature clusters; # = CXCR2 high clusters. (G) X-shift clusters 24 (red) and 23 (blue) displayed on UMAP plot (left) and density plots (right) comparing clusters 24 and 23 to remaining clusters for CD10 against CD66b and CD62L. (H) Distribution of clusters 23 and 24

### CXCR2-expressing neutrophils are associated with severe COVID-19

UMAP analysis confirmed the maintenance of CXCR2-expressing neutrophils in severe disease (Fig 2I). Interestingly, X-shift analysis identified additional clusters specific for severe COVID-19 (clusters 11, 12, 13, and 16) (Figs 2F and S5C). These clusters all showed elevated expression of CXCR2 (Fig 2J) and were notably absent in moderately ill patients (Figs 2K and S5C). On the other hand, the clusters were significantly elevated in severe disease (Fig 2K) and may therefore represent useful prognostic biomarkers. Such enrichment of CXCR2$^{hi}$ clusters suggests that neutrophils from severe patients may maintain responsiveness to chemokines such as IL-8.

### Partially activated immature neutrophils persist 12 wk after hospitalisation

We next asked whether there were any unique clusters associated with convalescence, as suggested by UMAP analysis (Fig 2B). We identified a single subpopulation (cluster 3) which was enriched in convalescent patients (Figs 2F and S5D). Interestingly, cluster 3 also showed reduced CD10 expression (Fig S5E), indicating that immature neutrophils persist up to 12 wk post-infection. This convalescent-enriched cluster was not associated with degranulation (CD66b$^{low}$) but did demonstrate partial activation, as evident from CD62L shedding (Fig S5E). When we stratified convalescent individuals into either outpatients or previously hospitalised patients, we found cluster 3 to be significantly enriched in individuals who had been admitted to the hospital because of COVID-19 complications (Fig S5F and G). Cluster 3 enrichment was not associated with either disease severity (Fig S5H) or total neutrophil counts upon admission or 12 wk post-hospitalisation (Fig S5I). In addition, we found no significant difference in absolute neutrophil count between acute infection and 12 wk post-initial hospitalisation (Fig S5J). This finding demonstrates that neutrophil dysregulation persists for prolonged periods after viral clearance, but further studies are needed to validate this result and investigate the clinical relevance of such a change.

## Discussion

Neutrophils were historically considered to be homogeneous cells, but recent studies have highlighted their phenotypic and functional plasticity (63). We hypothesized that divergent neutrophil developmental states may be important determinants of COVID-19, a disease that is characterised by a disordered myeloid response and a hyperinflammatory clinical phenotype (14, 64, 65).

Our study confirms multiple reports of expansion of neutrophil precursors in severe COVID-19 (30, 38, 40, 66). To test the functional consequences of this shift to immaturity, we used multiparameter cytometric analyses and showed that CD10$^-$ neutrophils are hyperactivated, with elevated exposure of secondary granule markers (CD66b and CD177) and proteolytic cleavage of the surface receptor CD62L (14). Thus, in addition to confirming previous associations between immature neutrophils and severe COVID-19, we provide functional data in support of the elevated proinflammatory potential of these cells.

Several published reports are consistent with our conclusions on neutrophil degranulation in COVID-19. MMP8 and other secondary granule components were found to be significantly elevated in the plasma of ICU versus non-ICU COVID-19 patients (13). Peripheral blood RNA sequencing has also identified a signature of neutrophil degranulation which distinguishes severe COVID-19 from mild disease (28, 30). Furthermore, multiple studies have highlighted CD177, a protein associated with secondary granules, as predictive of poor clinical outcome (16, 67). Secondary granules contain multiple proteins involved in cytoadhesion (68, 69), as well as metalloproteases with immunomodulatory and tissue degradation properties (69, 70), suggesting altered endothelial interactions by degranulated CD10$^{low}$ neutrophils. Furthermore, the contents of these granules also mediate interactions with platelets (71). Circulating CD10$^-$ neutrophils may thus specifically be related to immunothrombus formation and coagulopathy (72).

Our findings are consistent with reports of altered function of immature neutrophils in other diseases. Circulating CD10$^{low}$ neutrophils are thought to result from premature release from the bone marrow and have been detected in multiple diseases, including cancer (73) and bacterial sepsis (74). They are mobilised by the growth factor GCSF (44), which is elevated in COVID-19 patients (19). CD10$^-$ neutrophils typically sediment with "low-density granulocytes" and PBMCs on Ficoll density gradients (44, 75). In vasculitis, CD10$^-$ neutrophils have a reduced ability to generate NETs but a higher capacity to cause endothelial permeability, compared with CD10$^+$ neutrophils (76). Similarly, in systemic lupus erythematosus, immature CD10$^-$ neutrophils were found to have reduced capacity to phagocytose bacteria and produce NETs but had an elevated propensity to degranulate (75). In lung cancer, CD10$^{low}$ neutrophils correlate with advanced disease and also co-express PD-L1, raising the possibility that immature neutrophils may also have suppressive functions (73). These studies illustrate the need for deep phenotypic characterisation of CD10$^-$ cells with directed mechanistic studies to fully understand how this developmental shift impacts disease progression.

Unexpectedly, CD10$^{low}$ neutrophil subsets persist for at least 12 wk post-hospitalisation. It is interesting to speculate that persistent neutrophil alterations may contribute to "long COVID" symptoms, but additional, large-scale studies are needed to explore such a link. Furthermore, our study does not permit us to distinguish

---

among stratified groups (n = HC-20, mild 3, mod 14, sev 18, and conv 38). **(I)** UMAP plots stratified into healthy control and patient neutrophils with overlayed expression of CXCR2. **(J)** X-shift clusters 11, 12, 13, and 16 as displayed on UMAP plot (left), histogram demonstrating CXCR2 expression in clusters 11, 12, 13, 16, and remaining clusters (right). **(K)** Distribution of neutrophils from X-shift clusters 11, 12, 13, and 16 among stratified groups (n = HC-20, mild 3, mod 14, sev 18, and conv 38). **(G, J)** Statistical tests were as follows: (G, J) were analysed by the Kruskal–Wallis test with Dunn's multiple comparisons displayed on the graph. HC, healthy controls; Mod, moderate; Sev, severe; Conv, convalescent. Values represent mean ± SD.

between immature neutrophils arising because of SARS-Cov-2 infection or ones induced by comorbidities.

In contrast to CD10 expression, which was reduced on all neutrophils from COVID-19 patients, we detected severity-specific differences in CXCR2 expression. Moderate disease was accompanied by reduced CXCR2 surface abundance, whereas expression was maintained at baseline levels in severe patients. We propose that CXCR2 down-regulation may be an adaptive mechanism that prevents progression to severe disease. CXCR2 is known to be dynamically regulated, both during maturation and inflammation. Typically, CXCR2 expression is gradually increased during maturation (77) but can also be rapidly down-regulated by internalisation upon exposure to high concentrations of ligand (54). Because both moderate and severe COVID-19-associated neutrophils are immature, it is unlikely that a difference in maturation state between these groups explains the divergent expression of CXCR2. Furthermore, we find that the CXCR2 ligand, CXCL8, is similarly elevated in both moderate and severe patient plasma, suggesting that down-regulation is not mediated by ligand-induced receptor internalisation. Our results are in line with other studies that have identified both CD10$^{low}$ CXCR2$^{high}$ neutrophil subsets in severe COVID-19 by RNA sequencing (14) and reduced CXCR2 expression in moderately ill patients (78).

In mouse models of influenza, CXCR2 is required for homing of neutrophils to the lung (79), and treatment with a CXCR2 inhibitor reduced lung pathology (80). Down-regulation of CXCR2 in moderate COVID-19 may therefore be a protective, adaptive response that fails to occur in severe disease. Maintenance of neutrophil CXCR2 expression in severe COVID-19 may promote lung trafficking and associated tissue damage and perfusion defects. The role of IL-8/CXCR2 signalling in disease progression is supported by whole blood RNA expression profiling (81 *Preprint*) and an independent proteomic study (56), both of which report specific up-regulation of the pathway in severe COVID-19. Indeed, CXCR2 inhibitors appear to have potential as therapeutic agents in severe COVID-19 (82, 83).

Consistent with a role for CXCR2 signalling, our mass spectrometry results revealed up-regulation of pathways implicated in cell motility and protein synthesis, in addition to a strong type I interferon response. Our proteomic analysis also identified signatures of protein synthesis in neutrophils from severe COVID-19 patients, namely EIF2 signalling. Others have identified up-regulation of the related EIF4 pathway, similarly involved in protein translation, in severe COVID neutrophils (47). These findings suggest that neutrophil translational activity may be contributing to immunopathogenesis.

A limitation of our study was the sample size, where, particularly for proteomics, analysis of data after stratification by patient outcome was unfortunately not possible or underpowered. This risks the reduced detection of significant alterations to protein content because of the low number of patient samples. Additional in-depth proteomes or single-cell sequencing would help identify neutrophil-specific proteins that might be biomarkers of severe COVID-19.

In summary, we show that in COVID-19, immature neutrophils engage in elevated secondary granule release and surface protease activity, indicating that this neutrophil state may be excessively activated (Fig 3) and present for long periods post-infection. In addition, we find reduced CXCR2 expression on neutrophils from moderately ill patients, potentially revealing an adaptive change that limits trafficking into organs. Our study confirms neutrophil subpopulations as prognostic biomarkers of disease severity and supports a role for neutrophils in COVID-19 immunopathogenesis. Finally, as proposed by others, neutrophil precursors, the IL-8 signalling pathway, and in particular CXCR2, may represent therapeutic targets for severe COVID-19 (82, 83).

# Materials and Methods

### Human subjects and samples

Written informed consent was obtained from all patient and healthy donors, or from patients' family if patients were too unwell to consent. Samples were obtained under research ethics approval of the DISCOVER study (Diagnostic and Severity Markers of COVID-19 to Enable Rapid Triage, NHS REC 20/YH/0121) and Bristol BioBank (NHS REC 20/WA/0053). The convalescent patient cohort consisted of a mixture of recovered outpatient healthcare workers, recruited through Bristol BioBank, and DISCOVER patients, who were recalled for follow-up 12 wk after initial hospital admission. Donor demographics and clinical data are described in Table 1. Patients were stratified by disease severity as described in Arnold et al (41). Venous blood was collected in EDTA tubes (BD Biosciences).

### Flow cytometry

Staining was performed on whole blood. 100 μl of blood was washed in PBS and incubated with Zombie Aqua live/dead stain (product number 423101; BioLegend) at room temperature for 10 min. Samples were incubated with FC-Block (Human TruStain FcX, product number 422302; BioLegend) in staining buffer (PBS, 5 mM EDTA, 0.5% BSA), followed by addition of primary antibodies in staining buffer and incubation on ice for 20 min. Cells were washed and incubated with fluorophore-conjugated streptavidin on ice for 15 min. Erythrocytes were lysed with chilled ACK lysis buffer (product number A10492-01; Gibco), and cells were fixed in 4% PFA for 20 min. Cells were resuspended in staining buffer and analysed on a BD X20 Fortessa flow cytometer. Data were analysed using FlowJo (FlowJo, LLC). See Table S2 for flow cytometry reagents.

### UMAP and FlowSOM analysis

Dimensionality reduction was performed using concatenated live neutrophil populations from all stratified groups using 1,000 events per donor (see Supplemental Data 1 for gating strategy). UMAP plugin was used on the FlowJo software platform to generate dimensionality reduction plots using neutrophil surface markers CD66b, CD177, CD15, CD63, CXCR4, CD16, CD10, CXCR2, and CD62L and the following setting: Euclidean 15, nearest neighbours 15, minimum distance 0.5, and number of components 2. X-Shift clustering was

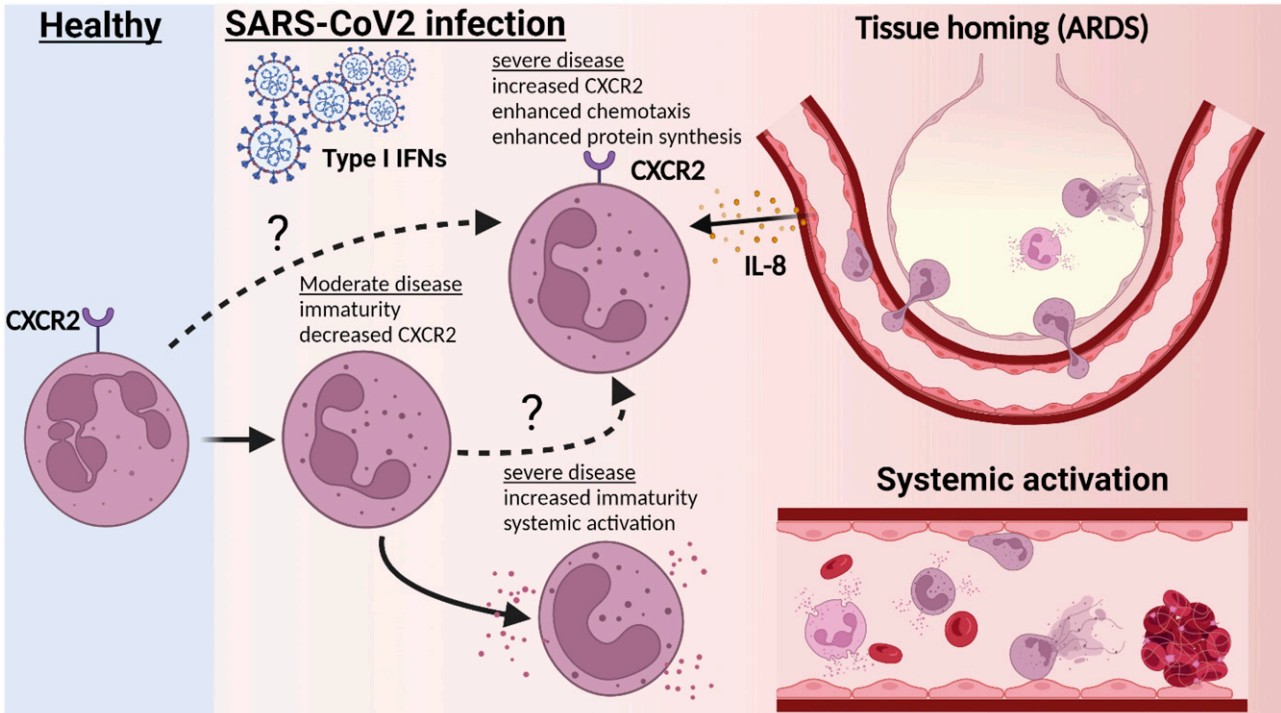

**Figure 3. Severe patient neutrophils fail to engage in protective reprogramming and instead engage in systemic activation.**
Upon acute infection with SARS-CoV-2, peripheral neutrophils become increasingly immature with reductions in markers associated with maturity, such as CD10. In moderately ill patients, this is accompanied by reductions in CXCR2 expression. In severe patients, CXCR2 is maintained in a subset of neutrophils. This potentially aids recruitment to infected lung tissue and may contribute to lung injury. In addition, CD10⁻ immature neutrophils in severe patients become activated in the periphery, with increased markers of degranulation and surface protease activity. These immature clusters lack CXCR2 expression, suggesting that instead of trafficking to inflamed organs, they may contribute to peripheral pathology, such as immunothrombosis. Created with BioRender.com.

generated using the FlowJo platform and used the same markers as UMAP, with the following settings: number of nearest neighbours (k): 20, distance metric: angular, and sampling limit: 94,000.

### CXCL8 measurements

CXCL8 was measured in plasma using the R&D Systems DuoSet Human IL-8/CXCL8 ELISA Kit (catalogue No. DY208) as per manufacturer's instructions.

### CXCR2 transcription

RNA was extracted from neutrophils using the PureLink RNA Mini Kit (catalogue number 12183025) from Thermo Fisher Scientific (Life Technologies) as per manufacturer's instructions. High-Capacity cDNA Reverse Transcription Kit (catalogue number 4368814) was used to convert 200 ng of RNA to cDNA as per manufacturer's instructions. CXCR2 transcription was analysed with 25 ng cDNA using Fast Sybr Green Master Mix from Applied Biosystems (catalogue number 4385612) and in-house-designed primers to CXCR2 and actin β as an endogenous control (Table S3). RT–PCR was performed using an Applied Biosystems QuantStudio 3 using the following parameters: 95°C for 20 s followed by 40 cycles of 95°C for 3 s followed by 60°C for 30 s. Melt curves were calculated to check if a single product was amplified, and RT, RNA, and cDNA controls confirmed primers were not amplifying contaminating DNA. Transcript abundance was calculated using ΔΔCT method.

### Proteomic analysis

Protein lysates were obtained from neutrophils isolated with using the Stem Cell Whole Blood Neutrophil Isolation Kit (Stemcell) as per manufacturer's instructions. $5 \times 10^6$ cells were washed in PBS and resuspended in RIPA buffer supplemented with 2× protease inhibitor cocktail (Calbiochem, 2 mM PMSF (Sigma-Aldrich), 1× Halt Phosphatase (Thermo Fisher Scientific), 50 mM TCEP (Sigma-Aldrich), and 10 mM EDTA. To limit potential biohazard risk, samples were heat inactivated at 57°C for 30 min before storing at −20°C. Protein was quantified using the Pierce BCA Protein Assay Kit (Thermo Fisher Scientific) and 100 μg of protein used for Tandem Mass Tag (TMT) mass spectrometry.

50 μg of each sample was digested with trypsin (1.25 μg; 37°C, overnight), labelled with TMT 11 plex reagents according to the manufacturer's protocol (Thermo Fisher Scientific), and the labelled samples were pooled. A 100 μg aliquot of the pooled sample was evaporated to dryness, resuspended in 5% formic acid, and then desalted using a SepPak cartridge according to the manufacturer's instructions (Waters). Eluate from the SepPak cartridge was again evaporated to dryness and resuspended in buffer A (20 mM ammonium hydroxide, pH 10) before fractionation by high pH reversed-phase chromatography using an UltiMate 3000 liquid

chromatography system (Thermo Fisher Scientific). In brief, the sample was loaded onto an XBridge BEH C18 Column (130 Å, 3.5 μm, 2.1 × 150 mm; Waters) in buffer A, and peptides were eluted with an increasing gradient of buffer B (20 mM ammonium hydroxide in acetonitrile, pH 10) from 0–95% over 60 min. The resulting fractions (15 in total) were evaporated to dryness and resuspended in 1% formic acid before analysis by nano-LC MSMS using an Orbitrap Fusion Tribrid mass spectrometer (Thermo Fisher Scientific).

High pH RP fractions were further fractionated using an Ultimate 3000 nano-LC system in line with an Orbitrap Fusion Tribrid mass spectrometer. In brief, peptides in 1% (vol/vol) formic acid were injected onto an Acclaim PepMap C18 nano-trap column (Thermo Fisher Scientific). After washing with 0.5% (vol/vol) acetonitrile, 0.1% (vol/vol) formic acid peptides were resolved on a 250 mm × 75 μm Acclaim PepMap C18 reverse phase analytical column (Thermo Fisher Scientific) over a 150-min organic gradient, using seven gradient segments (1–6% solvent B over 1 min, 6–15% B over 58 min, 15–32% B over 58 min, 32–40% B over 5 min, 40–90% B over 1 min, held at 90% B for 6 min, and then reduced to 1% B over 1 min) with a flow rate of 300 nl min$^{-1}$. Solvent A was 0.1% formic acid and solvent B was aqueous 80% acetonitrile in 0.1% formic acid. Peptides were ionized by nano-electrospray ionization at 2.0 kV using a stainless-steel emitter with an internal diameter of 30 μm (Thermo Fisher Scientific) and a capillary temperature of 275°C.

All spectra were acquired using an Orbitrap Fusion Tribrid mass spectrometer controlled by Xcalibur 2.1 software (Thermo Fisher Scientific) and operated in data-dependent acquisition mode using a synchronous precursor selection-MS3 workflow. FTMS1 spectra were collected at a resolution of 120,000, with an automatic gain control (AGC) target of 200,000 and a max injection time of 50 ms. Precursors were filtered with an intensity threshold of 5,000, according to charge state (to include charge states 2–7), and with monoisotopic peak determination set to peptide. Previously interrogated precursors were excluded using a dynamic window (60 s ± 10 ppm). The MS2 precursors were isolated with a quadrupole isolation window of 1.2 m/z. ITMS2 spectra were collected with an AGC target of 10,000, max injection time of 70 ms, and CID collision energy of 35%.

For FTMS3 analysis, the Orbitrap was operated at 50,000 resolution with an AGC target of 50,000 and a max injection time of 105 ms. Precursors were fragmented by high-energy collision dissociation at a normalised collision energy of 60% to ensure maximal TMT reporter ion yield. Synchronous precursor selection was enabled to include up to 10 MS2 fragment ions in the FTMS3 scan.

The raw data files were processed and quantified using Proteome Discoverer software v2.1 (Thermo Fisher Scientific) and searched against the UniProt Human Database (downloaded in January 2021) using the SEQUEST HT algorithm. Peptide precursor mass tolerance was set at 10 ppm, and MS/MS tolerance was set at 0.6 D. Search criteria included oxidation of methionine (+15.995 D), acetylation of the protein N-terminus (+42.011 D), and methionine loss plus acetylation of the protein N-terminus (−89.03 D) as variable modifications and carbamidomethylation of cysteine (+57.021 D) and the addition of the TMT (+229.163 D) to peptide N-termini and lysine as fixed modifications. Searches were performed with full tryptic digestion, and a maximum of two missed cleavages were allowed. The reverse database search option was enabled, and all data were filtered to satisfy false discovery rate of 5%.

## Bioinformatics analysis of proteomics

After analysis in Proteome Discoverer 2.1, the proteomics data were processed and further analysed in the R statistical computing environment. The protein groups were reassessed by an in-house script which selects a master protein firstly by ID and quantitation metrics, then by the annotation quality of uniprot accessions. Data were log$_2$ transformed, and statistical significance was calculated using Welch's $t$ test. IPA analysis was performed with a filter of $P < 0.05$ to identify biological trends in the proteins which are statistically significant between clinical conditions. PCAs were calculated using the PCA function in the FactoMineR package and plotted using either ggplot (2D) or Plotly (3D).

## Statistical analysis

Statistical analysis was performed using GraphPad Prism 8. Details of statistical tests used are detailed in the figure legend. Where n number was sufficient, normality was tested before analysis using the D'Agostino–Pearson test. For data assumed to be normally distributed (e.g., MFI), normality was improved with log transformation before analysis. Percentage data were assumed nonparametric, and appropriate tests were used. Asterisks on the graph represent the following: *$P < 0.05$, **$P < 0.01$, and ***$P < 0.001$. The absence of asterisk indicates non-significant data.

# Data Availability

Data are freely available upon request to the corresponding authors.

# Supplementary Information

# Acknowledgements

We thank Andrew Herman and Helen Rice for assistance with flow cytometry and Kate Heesom for proteomics. We also thank Marianna Santopaolo, Michaela Gregorova, and Lea Knezevic for help with sample processing, as well as Prof. Inge Hers, Jill King, Jane Metz, Charlie Plumptre, Begonia Morales-Aza, Lucy Collingwood, and Jenny Oliver for their work in assisting with participant recruitment and consent. Finally, we thank Alan Hedges for reviewing the statistical approaches used in the manuscript. B Amulic is funded by MRC grant MR/R02149X/1. L Rivino received a TRACK award from Elizabeth Blackwell Institute (EBI) for Health Research, University of Bristol, with funding from the University's alumni and friends.

## Author Contributions

CM Rice: conceptualization, data curation, formal analysis, investigation, methodology, and writing—original draft, review, and editing.

P Lewis: data curation, formal analysis, investigation, and writing—original draft.

FM Ponce-Garcia: investigation.

W Gibbs: investigation.

S Groves: investigation.

D Cela: investigation.

F Hamilton: resources.

D Arnold: resources and data curation.

C Hyams: resources.

E Oliver: resources and data curation.

R Barr: resources.

A Goenka: resources.

AD Davidson: resources.

L Wooldridge: resources and data curation.

A Finn: conceptualization and resources.

L Rivino: conceptualization and resources.

B Amulic: conceptualization, data curation, formal analysis, supervision, funding acquisition, investigation, project administration, and writing—original draft, review, and editing.

## Conflict of Interest Statement

The authors declare that they have no conflict of interest.

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
