## [Reviewer comments · Life Science Alliance]

Life Science Alliance

Hyperactive immature state and differential CXCR2 expression of neutrophils in severe COVID-19

Christopher Rice, Phillip Lewis, Fernando Ponce-Garcia, Willem Gibbs, Sarah Groves, Drinalda Cela, Fergus Hamilton, David Arnold, Catherine Hyams, Elizabeth Oliver, Rachael Barr, Anu Goenka, Andrew Davidson, Linda Wooldridge, Adam Finn, Laura Rivino, and Borko Amulic

DOI: <https://doi.org/10.26508/lsa.202201658>

Corresponding author(s): Borko Amulic, University of Bristol and Christopher Rice, University of Bristol

Review Timeline:

Submission Date:	2022-08-08
Editorial Decision:	2022-08-08
Revision Received:	2022-11-04
Editorial Decision:	2022-11-21
Revision Received:	2022-11-25
Accepted:	2022-11-28

Transaction Report:

Revision Plan

Manuscript number: RC- 2022-01450

Corresponding author(s): Borko Amulic

1. General Statements

The reviewers raised some useful points that will improve the manuscript.

Reviewer 1 questioned the novelty of our report (“*This work is interesting, yet the main problem of this work is that it lacks of novelty...*” In response, we would like to more clearly point out three novel contributions of our manuscript:

- 1) The reviewer is correct that CD10low, immature neutrophils have been linked to severe COVID-19 in multiple reports. It was precisely in response to these reports that we set out to investigate, in greater detail, the activation status of immature neutrophils. To the best of our knowledge, there are no reports showing that CD10low neutrophils are in fact engaging in elevated rates of secondary granule release (CD66b and CD177 upregulation) and elevated surface protease activity (CD62L cleavage), both of which are commonly used readouts of neutrophil activation. What has been previously reported for immature neutrophils is simply an association with severe COVID-19, however there is a lack of mechanistic understanding of how they may contribute to pathology. Our report builds on these previous publications and provides evidence that immature neutrophils display elevated degranulation rates (expose secondary granule markers on the surface), compared to mature neutrophils, in vivo. This finding is an important addition to our understanding of immature neutrophils and may help pave the way for targeting them therapeutically. We will state this more clearly in the revised manuscript.
- 2) Neutrophil CXCR2 surface expression has not been extensively investigated in COVID-19, despite the fact that this is the main chemotactic receptor, with obvious relevance to lung trafficking. Our report identifies a link between surface CXCR2 expression and disease severity: neutrophils from moderately ill patients downregulate, while neutrophils from severe COVID-19 patients maintain CXCR2 levels, potentially identifying a protective mechanism in moderate COVID-19, that is of clinical and therapeutic relevance. In fact, since submission of our manuscript, a Phase II clinical trial of the CXCR2 inhibitor Reparixin reported a significant decrease in adverse clinical outcomes in COVID-19, compared to standard of care (1). This further strengthens our hypothesis that modulation of CXCR2 is central to disease progression and emphasizes the need to publish this finding. We will state this more clearly in the revised manuscript.

3) Finally, we unexpectedly identified long-lasting, persistent changes in the neutrophil compartment, which have not been previously reported in disease (COVID-19 or others). We agree that our analysis is limited, and we will point out these limitations in the discussion, but we hope that this report spurs additional investigations into long term neutrophil changes, which may have relevance for long COVID and other inflammatory diseases.

(1) Landoni, G., Piemonti, L., Monforte, A.d. *et al.* A Multicenter Phase 2 Randomized Controlled Study on the Efficacy and Safety of Reparixin in the Treatment of Hospitalized Patients with COVID-19 Pneumonia. *Infect Dis Ther* (2022). <https://doi.org/10.1007/s40121-022-00644-6>

2. Description of the planned revisions

Major points:

1. The author 's main conclusions were based on flow cytometry. However, they didn't validate the purity of neutrophiles sorted by their sorting strategy.

We used an extensive flow cytometry panel to detect and analyse properties of neutrophils in whole blood with a high degree of precision. We did not sort the cells. We can provide a detailed validation of our gating strategy, and microscopy evidence that it does in fact identify neutrophils. This will be a useful addition to the manuscript.

2. The statical analysis should be checked by statisticians.

We can easily do this.

3. The author indicated they detected decreased expression of CD10 from moderate and severe COVID-19 patients, and concluded the potential of its prognostic utility. However, this conclusion is not novel, previous research performed by Silvin et al. and others have presented the immunosuppressive profile of CD10lowCD101-CXCR4+/- neutrophils in severe form of COVID-19 (PMID: 32810439, PMID: 33968405).

We completely agree with this comment. We replicated the association between immature neutrophils (CD10low) and disease. However we also go one step further and demonstrate two unique functional features of immature neutrophils: increased exposure of CD66b and CD177, indicative of secondary granule release (1), which contain important inflammatory and immunostimulatory factors and increased CD62L cleavage, which is indicative of elevated surface protease activity (2). To the best of our knowledge, this is the first demonstration of hyperactivation of immature neutrophils in any disease. We will state this clearly in the text.

Revision Plan

1) Lacy P. Mechanisms of degranulation in neutrophils. Allergy Asthma Clin Immunol. 2006 Sep 15;2(3):98-108. doi: 10.1186/1710-1492-2-3-98.

2) Li Y, Brazzell J, Herrera A, Walcheck B. ADAM17 deficiency by mature neutrophils has differential effects on L-selectin shedding. Blood. 2006 Oct 1;108(7):2275-9.

4. It seems that the author specifically picked CD10 to present its difference between patients and healthy controls, yet, for one thing the author didn't show how they detect the expression of CD10, did they perform western blotting, transcriptome or proteome? For another, the author did not show explain if CD10 is the only proteins or the top-ranked protein that show prognostic value.

The reviewer is correct that we specifically picked CD10. We were hypothesis driven and set out to examine activation state of CD10^{low}, immature neutrophils. We detected CD10 by a commercially available and well cited monoclonal flow cytometry antibody. This technique is superior to transcriptome and proteome because it quantifies surface expression of this receptor on an individual cell per cell basis, i.e allows us to quantify it in its physiological state. We further confirmed that we had indeed correctly detected immature neutrophils by use of a second surface receptor associated with neutrophil maturation CD101.

5. To further explore the neutrophil activation and chemotactic capacity, the author compared the proteomes of circulating neutrophils from severe and healthy controls. However, comparing to the published work, the sample numbers were too small, for there are only three severe patients enrolled, the author should include more samples for analysis.

Here we again agree with the reviewer: more extensive proteomic analyses have been reported by other groups. Our proteome data will be moved to the supplement because it does not contribute to the central findings of the manuscript (hyperactivation of immature neutrophils, reduced CXCR2 in moderate disease and long lasting neutrophil phenotypic changes). This will result in a more focused and streamlined manuscript.

6. The author performed UMAP analysis, and conclude long term perturbations to the myeloid compartments of convalescent patients. This conclusion is too rash, the author should include clinical index, such as absolute neutrophil counts, neutrophil percentage for integrative analysis.

We will include the data that the reviewer requests. The long term neutrophil phenotypic changes detected by UMAP are highly unexpected and novel but we agree with the reviewer that we did not perform an extensive analysis. We will change the wording in this section to make our conclusion more nuanced and less rash.

7. The proteins that the author indicated to be neutrophil functional related are more likely

to be functional universal. The author should include neutrophil specific datasets and screen out neutrophil specific markers for further analysis.

The proteome result will be moved to the supplement and we will include discussion of which proteins are neutrophils specific and which are likely to be universal. We wish to note here that the proteome was performed on isolated neutrophils, meaning all identified proteins are related to neutrophil function, but, quite rightly, perhaps not unique to neutrophils. We agree with the reviewer that it is interesting to discuss specific versus general activation pathways.

8. The author utilized X-Shift analysis to analyze the distinct neutrophil phenotypes in different disease states, yet, only one or two markers can hardly describe the whole picture. The author should conduct single cell transcriptome or proteome to systematically depict the diverse neutrophil phenotypes in different disease status.

We agree with the reviewer that flow cytometry is inherently limited because of restricted markers. However we wish to argue that the technique is also superior to other methods because it analyses protein abundance in the correct conformation and location (cell surface) and allows one to ask targeted questions. Furthermore, the X-shift analysis, was performed by comparing the expression of 10 surface markers simultaneously and categorising individual cells into automatically determined populations. Unfortunately, single cell transcriptomics or additional proteomes are beyond the scope of this manuscript.

9. There are multiple published papers describe the immune cell subsets of COVID-19 (PMID: 32838342, PMID: 33657410), the author should compare with them.

We will include a detailed discussion of how our findings compare to the above listed reports.

Minor point:

1. In table 1, the authors did not provide the p value among Mild, Moderate, and Severe groups.

We will perform statistical analyses on parameters in this table.

2. In Sup Fig 1B, Sup Fig 1C, Sup Fig 2E-G, I-K, Sup Fig 3D, the authors did not provide p value.

We will include this.

3. The author assumed "Principle component analysis (PCA) demonstrated heterogeneity amongst the severe patients, which was explained by patient outcome (Fig 2C)." Again, too small sample numbers, can hardly show the diversity.

We agree and will move this figure as well as other proteome data to supplement.

4. In Fig2G, the authors described patient neutrophils, and not described which type of patients.

Revision Plan

We will include this information.

5. The authors mentioned Fig1G in the sentence "Ingenuity pathway analysis (IPA) identified pathways related to chemotaxis, such as 'Signalling by Rho family GTPases', 'RhoA signalling' and 'Regulation of Actin-based Motility by Rho' as significantly enriched in patient neutrophils (Fig 2G), which aligns with maintained expression of CXCR2 (Fig 1G)", however we did not see the corresponding Fig1G.

We will change this figure to include the relevant data.

Reviewer #1 (Significance (Required)):

The paper lacks arguments regarding the novelty of the findings, as well as context with the current literature available for COVID-19 (several examples of the available literature references are provided) including comparison to published single cell dataset of COVID-19 (PMID: 32838342, PMID: 33657410, PMID: 32810439, PMID: 33968405). The paper focused more on known example, which are indeed useful to assess their strategy, but failed to detail their findings about unknown protein candidate which would bring more value to the manuscript.

As outlined above, the aim of our study was to analyse whether immature (CD10 low) neutrophils display altered functional properties, in order to elucidate their strong link to severe COVID 19. We did not set out to discover unknown protein candidates. We agree with the reviewer that it would be interesting to place our findings in the context of single cell transcriptome data, which would reveal whether our observed activated phenotype is mediated at the transcriptional or post transcriptional levels. We will carry out this analysis for the revision.

Reviewer #2 (Evidence, reproducibility and clarity (Required)):

The authors found that the expression of CXCR2 is decreased in patients with moderate COVID-19. However, the mechanisms were not explored.

CXCR2 downregulation is one of three findings in our manuscript. We will include a mechanistic experiment, which the reviewer suggests further down, to address why CXCR2 downregulation may be occurring (please see below). However, we wish to note that analysis of the activation status of immature neutrophils is also a major aim of the manuscript.

The hyperactivation status of neutrophils is not well defined, and proteomics data are not validated. The rationale for comparing healthy controls and severe COVID-19 patients is unclear. The manuscript in its current form raised more questions than answers.

We define hyperactivation as shedding of CD62L and surface exposure of granule membrane proteins. Both are standard readouts for neutrophil activation in vivo (1). The former measures activation of the surface protease ADAM17, which cleaves CD62L. The latter measures degranulation: fusion of granules with the plasma membrane leads to exposure of the receptors

Revision Plan

CD66b and CD177 (secondary granules) and CD63 (primary granules). Since immature neutrophils display multiple activation markers, and these are detected at higher levels than on mature neutrophils in the same sample, we have labelled this as hyperactivation.

1. Fortunati E, et al Human neutrophils switch to an activated phenotype after homing to the lung irrespective of inflammatory disease. Clin Exp Immunol. 2009 Mar;155(3):559-66.

We agree with the reviewer that the proteomics data are not central to our findings (beyond confirming that neutrophils are activated); we will move the data to the supplement. We agree with the reviewer that the term ‘hyperactivation’ is inappropriate when referring to the proteomics data, since there is no comparison with moderate COVID-19. We will remove this term and simply call it activation.

Major concerns:

1. No information is available on the healthy control group. How do they compare to the COVID-19 group? Age-, sex-differences? Comorbidities?

We will include these data.

2. Figure 1E. While the decrease in the level of CXCR2 expression in the moderate group is statistically significant, the functional significance of this finding is unclear. The MFI mean value of approximately five hundred units is still high. Whether it would be translated into decreased neutrophil migratory activity and tissue recruitment is unknown. As with any G-protein coupled receptor, the ligand-dependent stimulation of CXCR2 would induce its internalization. Do the authors consider the possibility of increased levels of CXCR2 ligands causing lower cell surface levels of CXCR2 in patients with moderate illness?

This is an interesting suggestion. We will measure circulating levels of IL-8, the major CXCR2 ligand, in plasma of moderate and severe patients.

With respect to the comment on MFI: since MFI is relative, it is difficult to know whether the ‘approximately five hundred units’ that we detected in moderate patients consists of meaningful expression or is simply background. We will include an FMO control (Fluorescence minus one) to answer this.

3. The proteomic analysis would be helpful in the identification of potential mechanisms involved in the reduced level of CXCR2 in the moderate group. However, the authors have decided to perform this analysis on healthy controls and patients with severe COVID-19 illness, two groups with a similar level of CXCR2 expression.

We agree with the reviewer that the mass spectrometry data is not appropriate for the mechanistic analysis. We will move the mass spec data to the supplement. We will address the mechanism of CXCR2 downregulation using stored patient samples:

- a) To test for transcriptional suppression, we will quantify CXCR2 mRNA abundance using qPCR in moderate and severe patients (paired with surface CXCR2 expression data).
- b) To test for internalisation of CXCR2, we will perform immunoblotting to detect internal CXCR in neutrophil lysates from patients where we detected reduced surface CXCR2 expression.

These two approaches are expected to begin to elucidate the mechanism by which CXCR2 is specifically reduced in moderately ill patients.

4. Figure 2. No information is available on the selection criteria for the samples used in proteomic analysis. How representative were those four healthy controls and three COVID-19 patients for their respective groups?

We will provide data on donor characteristics for healthy donors, as well as clinical data for patients.

5. Figure 2. It is unclear why the authors believe that the changes identified in proteomic analysis indicate the hyperactivation status of neutrophils. The analysis is performed by comparing neutrophils from the severe COVID-19 group against healthy control subjects. Would it be different for mild or moderate illness groups if compared to patients with severe illness or healthy subjects? Without these data, it is hard to understand if reported changes indicate hyperactivation.

We agree with the reviewer. The more appropriate term is ‘activation’, rather than ‘hyperactivation’, since proteomic analysis of moderate COVID is not available. We will change this term throughout the manuscript.

6. The authors' statement on neutrophil activation is not confirmed by any measurements in vitro or in vivo. It is unclear if these neutrophils produce more proinflammatory cytokines or reactive oxygen species? Are they more prone to undergo NETosis?

We disagree with the author here. We analysed neutrophil activation status ex vivo, using flow cytometry. As explained above, we tested surface protease activity as well as degranulation, both of which are important neutrophil functional responses.

To confirm our flow cytometry data and provide more evidence of elevated degranulation in patients, we measured degranulation in isolated neutrophils, treated with a strong inducer of degranulation (Calcium ionophore). We found significantly elevated release of secondary granule proteins OLFM, LYZ and LCN2, validating our flow cytometry data. We will include the data in the revised manuscript.

Minor:

7. It is unclear why the statistical approach in Figures 1A and B is different from the approach used in Figures 1C, D, and E.

Revision Plan

The differing approaches are due to the different nature of the data produced and their distribution. For percentile data, where we are detecting increases from negative (CD63 >0%) or reductions from fully positive (CD62L <100%), the data will be, by definition, unlikely to be normally distributed. Indeed, multiple normality tests on this data (Anderson-Darling, D'Agostino & Pearson, Shapiro-Wilk and Kolmogorov-Smirnov) demonstrated that the data were not normally distributed and therefore non-parametric tests (Kruskal Wallis with Dunn's multiple comparisons) were performed. For MFI data, due to the continuous nature of the data (i.e. not constrained by minimal, 0%, or maximal, 100%, values) likelihood of normal distribution is far higher. Normality tests found that most of the MFI data were normally distributed, however exceptions occurred in some groups. To improve the likelihood of normal distribution of the data, we performed log transformation of MFI data and analysed the log-transformed numbers with parametric tests such as One-way ANOVA with Tukey's multiple comparisons. We will confirm our approach with a statistician prior to resubmission.

8. Figure 1A, flow cytometric dot plot: It is interesting to see that the immature neutrophils are represented by a distinct subset of CD10- cells. In other studies, including those cited by the authors, immature neutrophils are characterized by gradually decreased expression of CD10, not distinctly separated from mature neutrophils.

We agree with the reviewer that this is interesting. It was observed in patients with severe COVID-19, although we also observed gradual, continuous decreases. We have also observed complete absence of CD10 in severe malaria and we think it is a feature of very immature neutrophils. We will discuss this further in the revised manuscript.

9. In Supplemental Figure 1 - the gating strategy for singlets is mislabeled; should be FSC-A vs. FSC-H, but listed as FSC-A vs. SSC-A.

We will make this change.

10. It may increase the translational value of the study if the authors perform an analysis of immune markers against clinical parameters demonstrating the severity of illness, e.g., hospital length of stay or hospital-free days, patients in an intensive care unit (ICU) versus non-ICU, and lab tests, serum CRP, WBC, NLR.

We will carry out this analysis for both CD10 and CXCR2.

Reviewer #2 (Significance (Required)):

In the current study, Rice et al. investigated the subpopulation of peripheral blood neutrophils obtained from patients with COVID-19 and healthy controls. The authors performed flow cytometric and proteomic analyses to determine the association between immunophenotype and activation of neutrophils and the severity of COVID-19 illness. The

Revision Plan

flow cytometric analysis is meticulously executed and informative and confirms previously published data on the immature status of circulating neutrophils in COVID-19.

We thank the reviewer for deeming our analysis ‘meticulously executed and informative’. We would like to point out that confirmation of ‘previously published data on the immature status of circulating neutrophils in COVID-19’ pertains to Figure 1A only. The remainder of the flow cytometry analysis aims to determine the activation status and pro-inflammatory potential of these cells, and we believe this analysis to be the first of its kind in COVID-19.

3. Description of the revisions that have already been incorporated in the transferred manuscript

4. Description of analyses that authors prefer not to carry out

We cannot carry out single cell RNA sequencing of neutrophils as suggested by reviewer 1, because this is beyond the scope of this manuscript.

August 8, 2022

Re: Life Science Alliance manuscript #LSA-2022-01658

Borko Amulic
University of Bristol
School of Cellular and Molecular Medicine,
University Walk
Bristol
United Kingdom

Dear Dr. Amulic,

Thank you for submitting your manuscript entitled "Hyperactive immature state and differential CXCR2 expression of neutrophils in severe COVID-19" to Life Science Alliance. We invite you to re-submit the manuscript, revised according to your Revision Plan.

Thank you for this interesting contribution to Life Science Alliance. We are looking forward to receiving your revised manuscript.

Sincerely,

B. MANUSCRIPT ORGANIZATION AND FORMATTING:

Referee #1

Evidence, reproducibility and clarity

Summary

In this manuscript, the authors used flow cytometry to investigate activity and phenotypic diversity of circulating neutrophils in acute and convalescent COVID-19 patients (acute COVID-19 patients: 34; healthy controls: 20). Further analysis indicated that hyperactivation of immature CD10⁻ subpopulations in severe disease. Additionally, the authors found CXCR2 was down-regulated in moderately ill patients, and CD10⁻ and CXCR2^{hi} neutrophil subpopulations were enriched in severe disease. This work is interesting, yet the main problem of this work is that it lacks of novelty, and the conclusion was proposed without solid evidence.

We thank the reviewer for their comment. In response, we would like to more clearly point out three novel contributions of our manuscript:

- 1) The reviewer is correct that CD10^{low}, immature neutrophils have been linked to severe COVID-19 in multiple reports. It was precisely in response to these reports that we set out to investigate, in greater detail, the activation status of immature neutrophils. To the best of our knowledge, there are no reports showing that CD10^{low} neutrophils are in fact engaging in a) elevated rates of secondary granule release (CD66b and CD177 externalisation) and b) elevated surface protease activity (CD62L cleavage), both of which are commonly used readouts of neutrophil activation. What has been previously reported for immature neutrophils is simply an association with severe COVID-19, and there is currently a lack of functional characterisation of how they may contribute to pathology. Our report builds on these previous publications and provides evidence that immature neutrophils display elevated degranulation rates compared to mature neutrophils, *in vivo*. This finding is an important addition to our understanding of immature neutrophils and may help pave the way for targeting them therapeutically. We have stated this more clearly in the manuscript, please see 'Discussion' lines 299-305.
- 2) Neutrophil CXCR2 surface expression has not been extensively investigated in COVID-19, despite the fact that this is the main neutrophil chemotactic receptor, with obvious relevance to lung trafficking. Our report identifies a link between surface CXCR2 expression and disease severity: neutrophils from moderately ill patients downregulate CXCR2, while neutrophils from severe COVID-19 patients maintain higher CXCR2 expression. We are therefore proposing a potential protective/adaptive mechanism in moderate COVID-19, which may act to limit neutrophil infiltration of organs in moderate disease. This finding is of clinical and therapeutic relevance. In fact, since submission of our manuscript, a Phase II clinical trial of the CXCR2 inhibitor Reparixin reported a significant decrease in adverse clinical outcomes in COVID-19, compared to standard of care (1). This further strengthens our hypothesis that modulation of CXCR2 is central to disease progression and emphasizes the need to publish this finding. We have stated this more clearly in the revised manuscript, please see 'Discussion', lines 337-341.
- 3) Finally, we unexpectedly identified long-lasting, persistent changes in the neutrophil compartment, which, to our knowledge, have not been previously reported in disease (neither COVID-19 nor in other infectious disease). We agree that our analysis is

limited and have pointed out these limitations in the discussion (lines 332-336); we hope that this report encourages additional investigations into long term neutrophil changes, which may have relevance for long COVID and other inflammatory diseases.

- (1) Landoni, G., Piemonti, L., Monforte, A.d. *et al.* A Multicenter Phase 2 Randomized Controlled Study on the Efficacy and Safety of Reparixin in the Treatment of Hospitalized Patients with COVID-19 Pneumonia. *Infect Dis Ther* (2022). <https://doi.org/10.1007/s40121-022-00644-6>

Major points:

1. The author's main conclusions were based on flow cytometry. However, they didn't validate the purity of neutrophils sorted by their sorting strategy.

We thank the reviewer for pointing out that our gating strategy was unclear. We used an extensive flow cytometry panel to detect and analyse properties of neutrophils in unsorted whole blood, with a high degree of precision. We have now included a detailed validation of our gating strategy, and microscopy evidence that it does in fact identify neutrophils. Please see Supplemental Figure 1 and lines 134-136. This is a valuable addition to the manuscript.

2. The statical analysis should be checked by statisticians.

As requested, we had the manuscript reviewed by a statistician, Dr. Alan Hedges (Honorary Research Fellow, University of Bristol), who found our use of statistics appropriate. The differing approaches are due to the distinctive nature of the data produced and their distributions. For percentile data, where we are detecting increases from negative (eg. CD63 >0%) or reductions from fully positive (eg. CD62L <100%), the data is, by definition, unlikely to be normally distributed. Indeed, multiple normality tests on this data (Anderson-Darling, D'Agostino & Pearson, Shapiro-Wilk and Kolmogorov-Smirnov) demonstrated that the data are not normally distributed and therefore non-parametric tests were performed (Kruskal Wallis with Dunn's multiple comparisons for more than 2 groups or Mann-Whitney tests for 2 parameters). For MFI data, due to the continuous nature of the data (i.e. not constrained by minimal, 0%, or maximal, 100%, values) the likelihood of normal distribution is far higher. Normality tests found that most of the MFI data were normally distributed, however exceptions occurred in some groups. To improve the likelihood of normal distribution of the data, we performed log transformation of MFI data prior to analysis with parametric tests (One-way ANOVA with Tukey's multiple comparisons for 3 or more groups and t-tests for comparing 2 groups). We have improved the wording of our statistical analysis in the methods section and in the figure legends throughout the manuscript.

3. The author indicated they detected decreased expression of CD10 from moderate and severe COVID-19 patients, and concluded the potential of its prognostic utility. However, this conclusion is not novel, previous research performed by Silvin et al. and others have presented the immunosuppressive profile of CD10lowCD101-CXCR4+/- neutrophils in severe form of COVID-19 (PMID: 32810439, PMID: 33968405).

We completely agree with this comment. We replicated the association between immature neutrophils (CD10low) and disease. However, we also go one step further and demonstrate two unique functional features of immature neutrophils: **a)** increased exposure of CD66b and CD177, indicative of secondary granule release (1), which contain important inflammatory and immunostimulatory factors and **b)** decreased CD62L cleavage, which is indicative of

elevated surface protease activity (2). To the best of our knowledge, this is the first demonstration of hyperactivation of immature CD10^{low} neutrophils compared to mature CD10^{hi} neutrophils, in COVID19. We have stated this more clearly in the text, please see lines 299-305.

1) Lacy P. Mechanisms of degranulation in neutrophils. *Allergy Asthma Clin Immunol*. 2006 Sep 15;2(3):98-108. doi: 10.1186/1710-1492-2-3-98.

2) Li Y, Brazzell J, Herrera A, Walcheck B. ADAM17 deficiency by mature neutrophils has differential effects on L-selectin shedding. *Blood*. 2006 Oct 1;108(7):2275-9.

4. It seems that the author specifically picked CD10 to present its difference between patients and healthy controls, yet, for one thing the author didn't show how they detect the expression of CD10, did they perform western blotting, transcriptome or proteome? For another, the author did not show explain if CD10 is the only proteins or the top-ranked protein that show prognostic value.

The reviewer is correct that we specifically picked CD10. We were hypothesis-driven and set out to investigate the activation state of CD10^{low}, immature neutrophils. We detected CD10 by a commercially available and well cited monoclonal flow cytometry antibody. This technique is superior to transcriptome and western blotting because it quantifies surface expression of this receptor on an individual cell-per-cell basis, i.e. allows us to quantify it in its physiological state. We further confirmed that we correctly detected immature neutrophils by use of a second surface receptor associated with neutrophil maturation (CD101).

5. To further explore the neutrophil activation and chemotactic capacity, the author compared the proteomes of circulating neutrophils from severe and healthy controls. However, comparing to the published work, the sample numbers were too small, for there are only three severe patients enrolled, the author should include more samples for analysis.

Here we again agree with the reviewer: more extensive proteomic analyses have been reported by other groups that are cited in the manuscript, and our findings have simply validated these reports. We have moved our proteome data to Supplementary Figure 4 because they are somewhat tangential to the central findings of the manuscript (hyperactivation of immature neutrophils, reduced CXCR2 in moderate disease and long-lasting neutrophil phenotypic changes). This resulted in a more focused and streamlined manuscript, so we thank the reviewer for this comment.

6. The author performed UMAP analysis, and conclude long term perturbations to the myeloid compartments of convalescent patients. This conclusion is too rash, the author should include clinical index, such as absolute neutrophil counts, neutrophil percentage for integrative analysis.

We thank the reviewer for this comment. The long term neutrophil phenotypic changes detected by UMAP are unexpected and novel but we agree with the reviewer that we did not perform an extensive analysis. We have changed the wording for this conclusion to make it more nuanced, please see lines 332-336.

As suggested we now include additional analysis pertaining to the presence of persistent immature neutrophil clusters in previously hospitalised convalescent patients. Firstly, we have additionally stratified percentage of neutrophils in cluster 3 by acute disease severity (Supplemental Fig 5H). Secondly, as suggested by the reviewers, we have correlated cluster

3 positivity against absolute neutrophil count from the acute infection or at 12 weeks post hospitalisation, when cluster 3 is detected (Supplemental Fig 5I). We additionally look for long term changes in neutrophil count by comparing absolute neutrophil count between acute infection and 12 weeks post hospitalisation in these patients (Supplemental Fig 5J). Unfortunately, none of these analyses identified a reciprocal alteration in total neutrophil counts and as discussed in lines 288-290, further investigation will be required to uncover the mechanism of long-term neutrophil alterations.

7. The proteins that the author indicated to be neutrophil functional related are more likely to be functional universal. The author should include neutrophil specific datasets and screen out neutrophil specific markers for further analysis.

We agree with the reviewer that the pathways that were detected by proteomic analysis (eg interferon response) are not neutrophil specific. However, we wish to note here that the mass spectrometry analysis was performed on isolated neutrophils, meaning all identified proteins are related to neutrophil function, but, as the reviewer points out, not exclusive to neutrophils. We agree with the reviewer that it is interesting to distinguish specific versus general activation pathways. The proteome result has been moved to the supplement (Supplemental Figure 4) and we have pointed out whether proteins are neutrophil specific or not in lines 224-225.

8. The author utilized X-Shift analysis to analyze the distinct neutrophil phenotypes in different disease states, yet, only one or two markers can hardly describe the whole picture. The author should conduct single cell transcriptome or proteome to systematically depict the diverse neutrophil phenotypes in different disease status.

We agree with the reviewer that flow cytometry is inherently limited because of restricted markers. However, we wish to argue that the technique is also superior to other methods because it analyses protein abundance in the correct conformation and location (cell surface) and allows one to ask targeted questions. Furthermore, the X-shift analysis was performed by comparing the expression of 10 surface markers simultaneously and categorising individual cells into automatically determined populations. Unfortunately, single cell transcriptomics or additional proteomes are beyond the scope of this manuscript, but we have included this point in the discussion (lines 371-373)

9. There are multiple published papers describe the immune cell subsets of COVID-19 (PMID: 32838342, PMID: 33657410), the author should compare with them.

We agree with the reviewer that there are multiple, comprehensive analyses looking at activation of immune cells in COVID-19 and that it is valuable to compare our neutrophil-specific analysis with these large-scale studies. We did this for the two reports mentioned above as well as 3 additional ones that we found. These are summarised below:

PMID: 32838342; Rodrigues et al 2020: this paper only reports elevated neutrophil counts in acutely ill patients and does not specifically analyse neutrophils. We added the reference to our introduction.

PMID: 33657410; Ren et al 2021: this paper performed RNA seq on the PBMC fraction, which normally excludes neutrophils and other granulocytes (except when neutrophil buoyancy is altered by activation). Interestingly, they also analysed bronchiolar lavage fluid (BALF) and found neutrophils to be enriched in the lungs of severely ill patients. The BALF neutrophils also had a strong type 1 interferon RNA signature, matching our proteomic findings. The authors also provide bioinformatic evidence that virus-RNA containing epithelial cells may be interacting with neutrophils. We have added this reference to the introduction and the discussion.

We also included more detailed comparison with the following reports:

Schulte-Schrepping et al, Severe COVID-19 Is Marked by a Dysregulated Myeloid Cell Compartment *Cell* 2020 Sep 17;182(6):1419-1440.e23.

This report uses RNA seq to identify accumulation of immature neutrophil subsets, as well as dysfunctional mature neutrophils in severe COVID-19. Although they do not specifically comment on activation status of CD10^{low} immature neutrophils, they do report that these cells are CXCR2^{high} in severe disease, which is reminiscent of our results. Furthermore, their immature clusters also demonstrate elevated transcription of secondary granule genes, which is consistent with our report of increased in vivo degranulation of secondary granule markers (CD66b and CD177). We have highlighted this in our discussion: lines 350-351

Silvin et al, Elevated Calprotectin and Abnormal Myeloid Cell Subsets Discriminate Severe from Mild COVID-19. *Cell*. 2020 Sep 17; 182(6): 1401–1418.e18.

This paper also analyses peripheral blood cells by RNA sequencing. They identify a specific neutrophil degranulation signature that distinguishes severe disease from mild disease, supporting our findings. We have included this reference (reference 30) in our discussion lines 309-310.

Lourda et al High-dimensional profiling reveals phenotypic heterogeneity and disease-specific alterations of granulocytes in COVID-19. *PNAS* 2021 Oct 5;118(40):e2109123118.

This was the only other study that we could find that analysed CXCR2 on neutrophils by flow cytometry. Similar to our findings, they report trends for reduced neutrophil CXCR2 on moderately ill patients and increased expression on neutrophils from severe patients. We included this reference in line 351.

Minor point:

1. In table 1, the authors did not provide the p value among Mild, Moderate, and Severe groups.

We thank the reviewer for pointing this out. We have now included p values for all continuous data. Please see updated Table 1 legend (Lines 545-549).

2. In Sup Fig 1B, Sup Fig 1C, Sup Fig 2E-G, I-K, Sup Fig 3D, the authors did not provide p value.

We thank the reviewer for pointing this out. The data that the reviewer refers to are non-significant. We have added information in the figure legend on the tests used to assess this, and clarified in the methods that the absence of asterisk indicates non-significant data (please see lines 503-509)

3. The author assumed "Principle component analysis (PCA) demonstrated heterogeneity amongst the severe patients, which was explained by patient outcome (Fig 2C)." Again, too small sample numbers, can hardly show the diversity.

We agree with the reviewer and have moved this figure as well as other proteome data to supplement figure 4. Discussion of low sample number is included in lines 368-371.

4. In Fig2G, the authors described patient neutrophils, and not described which type of patients.

The former Figure 2G (now supplementary Fig 4G) refers to patients analysed by proteomics. These were severe COVID-19 patients. We have additionally clarified this point in the figure legend (line 668) as well as in the main text (line 203).

5. The authors mentioned Fig1G in the sentence "Ingenuity pathway analysis (IPA) identified pathways related to chemotaxis, such as 'Signalling by Rho family GTPases', 'RhoA signalling' and 'Regulation of Actin-based Motility by Rho' as significantly enriched in patient neutrophils (Fig 2G), which aligns with maintained expression of CXCR2 (Fig 1G)", however we did not see the corresponding Fig1G.

We thank the reviewer for pointing out this mistake. We have corrected it: the correct figure in the revised manuscript is Fig 1E.

Reviewer #1 (Significance (Required)):

The paper lacks arguments regarding the novelty of the findings, as well as context with the current literature available for COVID-19 (several examples of the available literature references are provided) including comparison to published single cell dataset of COVID-19 (PMID: 32838342, PMID: 33657410, PMID: 32810439, PMID: 33968405). The paper focused more on known example, which are indeed useful to assess their strategy, but failed to detail their findings about unknown protein candidate which would bring more value to the manuscript.

As outlined above, the aim of our study was to analyse whether immature (CD10 low) neutrophils display altered functional properties, in order to elucidate their association with severe COVID 19. We did not set out to discover unknown protein candidates. We agree with the reviewer that it would be interesting to place our findings in the context of single cell transcriptome data, and we have updated the discussion to include these references: **PMID: 32838342; Rodrigues et al 2020** has been added to the introduction, as explained above (line 96).

PMID: 33657410; Ren et al 2021 confirms our findings on interferon response genes, as explained above, and has been added to the discussion (lines 99, 214 and 310).

PMID: 32810439; Silvin et al report a degranulation signature in severely ill patients, which matches our flow cytometry data on CD66b and CD177 exposure. This paper is discussed in lines 300 and 310.

Reviewer #2 (Evidence, reproducibility and clarity (Required)):

The authors found that the expression of CXCR2 is decreased in patients with moderate COVID-19. However, the mechanisms were not explored.

We thank the reviewer for pointing out that additional mechanistic details would be interesting. To address the reviewer's comment, we tested the hypothesis that CXCR2 downregulation occurs at the transcriptional level. We analysed CXCR2 transcription in neutrophils isolated from a limited number of healthy donors (n=3) and moderately ill COVID-19 patients (n=4), using quantitative RT-PCR. We found reduced CXCR2 mRNA abundance in patients, suggesting that downregulation of this receptor was occurring at the transcriptional level. The data has been included in Supplementary figure Sup Fig 3F. However we also point out the small sample size for this experiment and stress the need for confirmatory studies (lines 191-193).

The hyperactivation status of neutrophils is not well defined, and proteomics data are not validated. The rationale for comparing healthy controls and severe COVID-19 patients is unclear. The manuscript in its current form raised more questions than answers.

We define activation as shedding of CD62L and surface exposure of granule membrane proteins. Both are standard readouts for neutrophil activation *in vivo* (1). The former measures activation of the surface protease ADAM17, which cleaves CD62L. The latter measures degranulation: fusion of granules with the plasma membrane leads to exposure of the receptors CD66b and CD177 (secondary granules) and CD63 (primary granules). In response to the reviewer's comment, we have altered the text to only use the term 'hyperactivation' to refer to the elevated activation of CD10^{low} immature neutrophils, since these display higher levels of activation markers compared to mature neutrophils in the same sample.

We agree with the reviewer that the proteomics data are not central to our findings (beyond confirming that neutrophils are activated); we have moved the data to the supplement. We agree with the reviewer that the term 'hyperactivation' is inappropriate when referring to the proteomics data, since there is no comparison with moderate COVID-19. We have removed this term and simply refer to activation.

1. Fortunati E, et al Human neutrophils switch to an activated phenotype after homing to the lung irrespective of inflammatory disease. Clin Exp Immunol. 2009 Mar;155(3):559-66.

Major concerns:

1. No information is available on the healthy control group. How do they compare to the COVID-19 group? Age-, sex-differences? Comorbidities?

We thank the reviewer for pointing out this oversight. We have included demographic data for healthy controls in Table 2

2. Figure 1E. While the decrease in the level of CXCR2 expression in the moderate group is statistically significant, the functional significance of this finding is unclear. The MFI mean value of approximately five hundred units is still high. Whether it would be translated into decreased neutrophil migratory activity and tissue recruitment is unknown. As with any G-protein coupled receptor, the ligand-dependent stimulation of CXCR2 would induce its internalization. Do the authors consider the possibility of increased levels of CXCR2 ligands causing lower cell surface levels of CXCR2 in patients with moderate illness?

We thank the reviewer for the interesting suggestion on ligand dependent CXCR2 internalisation. This hypothesis predicts that circulating CXCR2 ligands would be more abundant in patients with moderate disease, compared to severe disease. In order to test this, we measured circulating levels of CXCL8 (IL-8), the major CXCR2 ligand (2), in plasma of moderate and severe patients. We found no significant difference in circulating CXCL8 in moderate and severe COVID19, with a trend towards increased concentration in severe disease (Supplemental Figure 3D), which is similar to published reports (3). We also directly correlated CXCL8 plasma concentration with CXCR2 MFI in a subset of patients and found no significant relationship between the two, suggesting that CXCR2 downregulation was not mediated by CXCL8 signaling.

With respect to the comment on MFI: since MFI is relative, it is difficult to know whether the 'approximately five hundred units' that we detected in moderate patients consists of meaningful expression or is simply background. We have included an FMO control ('Fluorescence minus one' control; Supplemental figure 3A), which suggests that despite significant reductions in expression, these neutrophils were not CXCR2 negative, possibly indicating there are still responsive to CXCR2 ligands. We updated the results section to include this information (lines 180-183)

2. Mukaida N. Pathophysiological roles of interleukin-8/CXCL8 in pulmonary diseases. *Am J Physiol Lung Cell Mol Physiol* 284: L566–L577, 2003
3. Kaiser et al. Self-sustaining IL-8 loops drive a prothrombotic neutrophil phenotype in severe COVID-19. *JCI Insight*. 2021;6(18):e150862.

3. The proteomic analysis would be helpful in the identification of potential mechanisms involved in the reduced level of CXCR2 in the moderate group. However, the authors have decided to perform this analysis on healthy controls and patients with severe COVID-19 illness, two groups with a similar level of CXCR2 expression.

We agree with the reviewer that the mass spectrometry data is not ideal for the mechanistic analysis. We have moved the mass spec data to the supplement. We have addressed the mechanism of CXCR2 downregulation using stored patient samples and shown that this appears to occur at the transcriptional level (please see above).

4. Figure 2. No information is available on the selection criteria for the samples used in proteomic analysis. How representative were those four healthy controls and three COVID-19 patients for their respective groups?

We thank the reviewer for pointing out this oversight and have included the demographics of healthy controls and severe COVID-19 patients used in our proteomics analysis in a new table (Supplemental table 1).

5. Figure 2. It is unclear why the authors believe that the changes identified in proteomic analysis indicate the hyperactivation status of neutrophils. The analysis is performed by comparing neutrophils from the severe COVID-19 group against healthy control subjects. Would it be different for mild or moderate illness groups if compared to patients with severe illness or healthy subjects? Without these data, it is hard to understand if reported changes indicate hyperactivation.

We agree with the reviewer. The more appropriate term is 'activation', rather than 'hyperactivation', since proteomic analysis of moderate patients is not available. We have changed this term in regard to the proteomics results (line 223).

6. The authors' statement on neutrophil activation is not confirmed by any measurements in vitro or in vivo. It is unclear if these neutrophils produce more proinflammatory cytokines or reactive oxygen species? Are they more prone to undergo NETosis?

We agree with the reviewer that it would be useful to have additional data but our study focused on two *ex vivo* activation readouts: degranulation status (primary and secondary granule markers) and surface protease activation (CD62L cleavage), both of which are important neutrophil functional responses. Since these FACS assays were carried out

immediately after blood draw, without additional purification or *in vitro* stimulation, this is the closest one can get to assaying neutrophil activity *in vivo*. Furthermore, we were interested in comparing activation of CD10^{low} and CD10^{hi} neutrophils, for which our flow-based assays are most appropriate. The reviewer's comment is valuable, and we have included references for reports where NETosis and ROS production were reported to be elevated in patients compared to healthy controls. Please see lines 102-104.

Minor:

7. It is unclear why the statistical approach in Figures 1A and B is different from the approach used in Figures 1C, D, and E.

The differing approaches are due to the different nature of the data produced and their distribution. For percentile data, where we are detecting increases from negative (CD63 >0%) or reductions from fully positive (CD62L <100%), the data will be, by definition, unlikely to be normally distributed. Indeed, multiple normality tests on this data (Anderson-Darling, D'Agostino & Pearson, Shapiro-Wilk and Kolmogorov-Smirnov) demonstrated that the data were not normally distributed and therefore non-parametric tests were performed (Kruskal Wallis with Dunn's multiple comparisons). For MFI data, due to the continuous nature of the data (i.e. not constrained by minimal, 0%, or maximal, 100%, values) likelihood of normal distribution is far higher. Normality tests found that most of the MFI data were normally distributed, however exceptions occurred in some groups. To improve the likelihood of normal distribution of the data, we performed log transformation of MFI data and analysed the log-transformed numbers with parametric tests such as One-way ANOVA with Tukey's multiple comparisons. We have confirmed the validity of our approach with a statistician prior to resubmission.

8. Figure 1A, flow cytometric dot plot: It is interesting to see that the immature neutrophils are represented by a distinct subset of CD10⁻ cells. In other studies, including those cited by the authors, immature neutrophils are characterized by gradually decreased expression of CD10, not distinctly separated from mature neutrophils.

We agree with the reviewer that this is interesting. It was observed in a subset of patients with severe COVID-19, although we also observed gradual, continuous decreases. We similarly observed complete absence of CD10 in another study on severe malaria and we think it is a feature of very immature neutrophils.

We have added a second FACS plot from an additional severe COVID-19 patient, to illustrate the variation in CD10 expression (Figure 1A):

9. In Supplemental Figure 1 - the gating strategy for singlets is mislabeled; should be FSC-A vs. FSC-H, but listed as FSC-A vs. SSC-A.

We thank the reviewer for spotting this mistake. We made the change.

10. It may increase the translational value of the study if the authors perform an analysis of immune markers against clinical parameters demonstrating the severity of illness, e.g., hospital length of stay or hospital-free days, patients in an intensive care unit (ICU) versus non-ICU, and lab tests, serum CRP, WBC, NLR.

We thank the reviewer for this suggestion: we performed correlative analysis between neutrophil CD10 expression or CXCR2 expression against a variety of clinical parameters including CRP, NLR and WBC count and the separation of patients based on care setting (intensive/high care versus non-ITU). While we found no correlation between either parameter and CRP, NLR and WBC count, we did find significantly higher proportion of CD10 negative neutrophils in intensive/high care, as well as a trend for increased CXCR2 in the same settings ($p=0.0601$). We have now included this data in the supplemental figures 2D and 3C.

Reviewer #2 (Significance (Required)):

In the current study, Rice et al. investigated the subpopulation of peripheral blood neutrophils obtained from patients with COVID-19 and healthy controls. The authors performed flow cytometric and proteomic analyses to determine the association between immunophenotype and activation of neutrophils and the severity of COVID-19 illness. The flow cytometric analysis is meticulously executed and informative and confirms previously published data on the immature status of circulating neutrophils in COVID-19.

We thank the reviewer for deeming our analysis 'meticulously executed and informative'. We would like to point out that confirmation of 'previously published data on the immature status of circulating neutrophils in COVID-19' pertains to Figure 1A only. The remainder of the flow cytometry analysis aims to determine the activation status and pro-inflammatory potential of these cells, which is essential for understanding their role in disease.

November 21, 2022

RE: Life Science Alliance Manuscript #LSA-2022-01658R

Dr. Borko Amulic
University of Bristol
School of Cellular and Molecular Medicine,
Biomedical sciences building
University Walk
Bristol BS8 1TD
United Kingdom

Dear Dr. Amulic,

Thank you for submitting your revised manuscript entitled "Hyperactive immature state and differential CXCR2 expression of neutrophils in severe COVID-19". We would be happy to publish your paper in Life Science Alliance pending final revisions necessary to meet our formatting guidelines.

- please add ORCID ID for secondary corresponding author-they should have received instructions on how to do so
- please add an abstract and a category for your manuscript to our system
- please use the [10 author names, et al.] format in your references (i.e. limit the author names to the first 10)
- please add a callout for Figure S4C to your main manuscript text

A. FINAL FILES:

B. MANUSCRIPT ORGANIZATION AND FORMATTING:

**Submission of a paper that does not conform to Life Science Alliance guidelines will delay the acceptance of your

manuscript.**

The license to publish form must be signed before your manuscript can be sent to production. A link to the electronic license to publish form will be sent to the corresponding author only. Please take a moment to check your funder requirements.

Sincerely,

Reviewer #1 (Comments to the Authors (Required)):

All concerns were appropriately addressed

Reviewer #2 (Comments to the Authors (Required)):

The authors have adequately addressed all my comments.

November 28, 2022

RE: Life Science Alliance Manuscript #LSA-2022-01658RR

Dr. Borko Amulic
University of Bristol
School of Cellular and Molecular Medicine,
Biomedical sciences building
University Walk
Bristol BS8 1TD
United Kingdom

Dear Dr. Amulic,

Thank you for submitting your Research Article entitled "Hyperactive immature state and differential CXCR2 expression of neutrophils in severe COVID-19". It is a pleasure to let you know that your manuscript is now accepted for publication in Life Science Alliance. Congratulations on this interesting work.

DISTRIBUTION OF MATERIALS:

Again, congratulations on a very nice paper. I hope you found the review process to be constructive and are pleased with how the manuscript was handled editorially. We look forward to future exciting submissions from your lab.

Sincerely,
